# Understanding the experience of veterans who require lower limb amputation in the veterans health administration

Chelsea Leonard[1]*, George Sayre[2,3,4,5], Sienna Williams[2,6], Alison Henderson[2], Daniel Norvell[2,6,7], Aaron P. Turner[2], Joseph Czerniecki[2,6,7]

1 Denver Seattle COIN. VA Eastern Colorado Healthcare System, Aurora, Colorado, United States of America, 2 VA Puget Sound Health Care System, Seattle, Washington, United States of America, 3 Qualitative Research Core, HSR&D Center of Innovation for Veteran-Centered and Value-Driven Care, Seattle, Washington, United States of America, 4 VA Collaborative Evaluation Center (VACE), Seattle, Washington, United States of America, 5 Department of Health Services, University of Washington, Seattle, Washington, United States of America, 6 VA Center for Limb Loss and Mobility (CLiMB), Seattle, Washington, United States of America, 7 Department of Rehabilitation Medicine, University of Washington, Seattle, Washington, United States of America

* Chelsea.Leonard@va.gov

**Data Availability Statement:** This paper reports on a qualitative study and excerpts of appropriate data are shared within the paper. We are unable to provide full de-identified interview transcripts as

## Abstract

### Purpose

There is limited qualitative research on the experience of patients undergoing lower limb amputation due to chronic limb threatening ischemia (CLTI) and their participation in amputation-level decisions. This study was performed to understand patient lived experiences related to amputation and patient involvement in shared decision making.

### Materials and methods

Phenomenological interviews were conducted with Veterans 6–12 months post transtibial or transmetatarsal amputation due to CLTI. Interviews were read and summarized by two analysts who discussed the contents of each interview and relationships between interviews to identify emergent, cross-cutting elements of patient experience.

### Results

Twelve patients were interviewed between March and August 2019. Three cross cutting elements of patient lived experience and participation in shared decision making were identified: 1) Lacking a sense of decision making; 2) Actively working towards recovery as response to a perceived loss of independence; and 3) Experiencing amputation as a Veteran.

### Conclusions

Patients did not report a high level of involvement in shared decision making about their amputation or amputation level. Understanding patient experiences and priorities is crucial to supporting shared decision making for Veterans with amputation due to CLTI.

this was not discussed with participants during the consent process.

**Funding:** Role of Funder: This material is based upon work supported by the US Department of Veterans Affairs, Office of Research and Development, Rehabilitation Research and Development Grant number 1 I01 RX002960-01. The funders had no role in study design, data collection and analysis, decision to publish, or preparation of the manuscript.

**Competing interests:** The authors have declared that no competing interests exist.

## Introduction

Lower limb amputation is a life changing and emotional event [1,2]. Approximately 150,000 patients in the United States receive lower limb amputations annually [3], and 10 percent of these amputations are performed in the Veterans Health Administration (VA) [4,5]. The rates of lower extremity amputation due to chronic limb threatening ischemia (CLTI) in the VHA are substantially higher than in the US population in general [6,7] and the absolute number of amputations in the VA is increasing, due to an increase in the Veteran population [8] and the increasing prevalence of diabetes and arterial vascular disease [9]. Between 2008 and 2013, the VA implemented the Amputation System of Care (ASoC) to enhance the quality of care provided to Veterans with limb loss [10].

The Department of Health and Human Services and the Institute of Medicine report of 2001 emphasized the importance of incorporating patient priorities in key health care decisions through shared decision making (SDM). Amputation level decisions for lower limb amputations present a unique opportunity to engage in SDM due to clinical uncertainty in amputation level recommendations. Published guidance on amputation level decisions for lower limb amputations is vague and indicates that surgeon and patient perceptions of the risk benefit ratio should be considered [11]. This lack of clinical guidance results in significant variation in amputation level selection in different health care systems, as well as regional variation within the VA [8,12–14]. Additional variation has been noted across gender and racial subgroups [13]. Incorporating patient wishes into these decisions is critical to promoting patient centered care.

Previous qualitative research on patient experiences with lower limb amputation indicates that patients wish to have an active role in the decision to amputate (1). However, patients report several barriers to participation in amputation decisions. They report receiving insufficient information related to recovery, rehabilitation, and prosthetics [15,16], and may have trouble assimilating information about amputation due to the overwhelming nature of the situation and technical language used to convey information [15]. It is currently unclear the degree to which patients feel involved in decision making, and how to best provide them with necessary information to facilitate their involvement.

Understanding if and how patients participate in SDM is critical to providing patient centered care and promoting patient participation in a surgical decision that has major implications for quality of life [15]. In this paper we describe elements of lived experience among lower limb amputees in the VA. The findings will provide important context in the development of tools and guidelines that will help include patients in the amputation level shared decision-making process.

## Methods

### Design

This work was completed as the first step of a larger study aimed at developing a patient decision aid and provider decision support tool. The goal of this portion of the study was to understand patient experiences around amputation and amputation level decisions and understand if SDM was taking place to assist in the development of these tools. An interpretive phenomenology approach (IPA) was chosen to allow participants to describe the salient aspects of their experience without researcher-imposed priorities.

### Participant selection and recruitment

Veterans eligible for interviews were 6–12 months post transtibial or transmetatarsal amputation due to CLTI and over 40 years old. Eligible patients were interviewed between March and

August 2019. Potential participants were identified in the Veteran's Affairs Corporate Data Warehouse which includes inpatient and outpatient data as well as demographic and contact information (Table 1).

Potential participants were mailed a recruitment letter and study information including an information statement with consent information. Those who did not respond to the mailing within 14 days were contacted two additional times by telephone. The recruitment letter included an "opt out" postcard and phone number. Interested individuals were screened over the phone to ensure that they met study criteria (SW). Informed consent was conducted by telephone prior to interviews; the interviewer reviewed the information statement with the participant during the consent conversation. Consent was recorded and witness by the interviewer as approved by the institutional review board.

### Interviews

Semi-structured lifeworld interviews as described by Brinkman and Kvale [17] were conducted to elicit Veterans' rich descriptions of their lived experiences. An interview guide was developed by the study team (GS, SW, CL, AH, DN, JC) consisting of open-ended questions and semi-structured probes [18] designed to facilitate participant descriptions of their everyday experience regarding amputation and amputation level decisions (S1 File). Interviews were conducted over the telephone by a trained interviewer (SW), recorded, and transcribed verbatim.

### Analysis

Interviews were analyzed using an interpretive phenomenological approach [19]. Interviews were read multiple times by two analysts (CL and SW) to obtain an overall sense of

**Table 1. Veteran interview participant demographics.**

| | N | % |
|---|---|---|
| **Amputation Level** | | |
| TT w/ or w/o revision | 3 | 25 |
| TM no revision | 4 | 33.3 |
| TM w/any revision | 5 | 41.7 |
| **VA Region** | | |
| North Atlantic | 4 | 33.3 |
| Southeast | 2 | 16.7 |
| Midwest | 1 | 8.3 |
| Continental | 4 | 33.3 |
| Pacific | 1 | 8.3 |
| **Age Category** | | |
| 40–49 | 0 | 0 |
| 50–59 | 4 | 33.3 |
| 60–69 | 6 | 50 |
| 70–79 | 1 | 8.3 |
| 80–89 | 1 | 8.3 |
| **Gender** | | |
| Male | 12 | 100 |
| **Race** | | |
| White | 6 | 50 |
| Black | 5 | 41.7 |
| Latino | 1 | 8.3 |

participants descriptions of experience. Analysts practiced bracketing to identify biases, recorded their biases prior to reading each interview and continued this process throughout the course of analysis [19,20]. Each analyst wrote summaries of each interview that included the most salient points made by the participant and the analyst's reaction to the interview. The analysts and methodologist (GS) discussed the contents of each interview and relationships between interviews to tentatively identify emergent, cross-cutting elements of patient experiences. These conversations extended the bracketing process by including a discussion of biases, reactions to the data, and how biases may have contributed to analysts' interpretation of interview content. Themes were then organized to develop a consistent and meaningful description of the meaning of participants' experience and each element of patient experience were summarized using key quotes from the interviews.

This study was reviewed and approved by VA Central IRB and Human Research Protection Program, VA Puget Sound Health Care System (MIRB 01700; IRBNet # 1587998–12).

## Results

Twelve interviews were conducted between March and August 2019. Interviews lasted between 45 and 90 minutes. Participant characteristics are summarized in Table 1. Participant descriptors to contextualize quotes are presented in Table 2. Three cross cutting elements of patient experience related to amputation and participation in SDM were identified: 1) Lacking a sense of decision making; 2) Actively working towards recovery as response to a perceived loss of independence; and 3) Experiencing amputation as a Veteran.

### Lacking a sense of decision making

When asked about the decision around amputation level, most participants described a passive role in decision making,

**Table 2. Participant information for quotations.**

| Patient # | Clinical Setting^β | Age Range° | Amputation Level* |
|---|---|---|---|
| 1 | ER | 3 | 2. TM |
| 2 | ER | 2 | 2. TM |
| 3 | ER | 2 | 1. TM |
| 4 | (ER) | 1 | 1. TT |
| 5 | (clinical) | 4 | 1. TT |
| 6 | ER | 1 | 2. TM |
| 7 | (clinical) | 1 | 1. TM |
| 8 | ER | 1 | 2. TM |
| 9 | ER | 2 | 2. TM |
| 10 | (ER) | 2 | 2. TM |
| 11 | (clinical) | 2 | 2. TM |
| 12 | (ER) | 2 | 3. TT-G |

^βCare setting was deduced from interview. Parentheses indicate instances where participant did not explicitly name care setting, but contextual clues pointed to either a clinical or ER setting.

°1 = 50–59; 2 = 60–69; 3 = 70–79; 4 = 80–89.

*TM = transmetatarsal, TT = transtibial, TT-G = transtibial guillotine.

*"Well, at first we tried to save the toe that they said was infected with gangrene. They did a radiation study, and did treatment, but it wasn't successful, so they decided to amputate. [. . .] when I say they, the doctors explained to me that with gangrene, if you don't contain it, it'll spread. So I didn't want it to spread. So I agreed to the amputation.." -Participant 5*

They discussed the medical necessity of amputation and did not perceive a choice in whether or where to amputate,

*"I had gangrene. What choice did she have? But she had to do exploratory surgery. I've got pictures of it. She opened up the bottom of the foot, and opened up the top. I wasn't going to make it until she got that gangrene out of there, because gangrene will spread like wildfire."— Participant 2*

Many participants expressed trust in their doctor or medical team's decision,

*"No, he just told me that it's a procedure that has to be done. He said he wished he couldn't do it, but that it had to be done. That was just the bottom line [. . .] \ I don't know, he just said that everything would be ok. So I just took his word for it and he did it."—Participant 3*

Others described agreeing to amputation,

*"Well, they tried hard, for at least a week and a half to save the toe. But, at a point after that, we decided, well, they decided that the toe was completely gone, I had no feeling in it. So that's when we decided to amputate. And that's where we are right now. I agreed to everything, it was nothing that was forced on me. They told me their findings about gangrene, and I knew the toe had no feeling in it, and that the toe was gone, so I went ahead and agreed to have an amputation. To have the amputation."- Participant 5.*

Most participants indicated that they did not have in-depth conversations with their doctor or surgeon prior to amputation,

*"But there was no real conversation about amputating my foot. I think when I told her it was ok if the doctors took my foot off, she got up to go tell them and I think she was perhaps looking for a tactful way of telling me that my foot had to come off [. . .]But no, there was no real conversation about having to amputate my feet."—Participant 12*

Notably, a few participants described amputations that took place while they were sedated with little to no prior conversation,

*"I don't know much about it, I was asleep. I was asleep, but when I woke up, all my toes were gone. And I didn't feel a thing, I didn't feel nothing"—Participant 11*

One described frustration related to his lack of choice,

*"It didn't feel very good, I mean, not having a choice. Like I said, they went through and they did all of the tests as far as looking for veins, and there weren't any. And I had already tried a cadaver vein, that didn't work out but maybe 18 months, and with all of that pain you had to go through, it wasn't worth it.."—Participant 1*

A few participants provided input on amputation level decisions. For example, one participant described the choice to amputate all of his toes in order to avoid a future lower leg amputation,

*"Yes, because, I'd rather have all of my toes, but if it came right down to it, I can function with no toes. I've adapted, adjusted and overcame. It would be a whole different thought process if they turned around and said, 'we hate to tell you this, but you've got an infection that's already gone down all the way into the foot, we're going to have to go below the knee and you're going to be missing below the knee'. If they say that it's going to have to come off, it's not too much of a choice, if you've got gangrene or if you've got something that's definitely wrong. [. . .] So, am I going to say that I would not? I would listen to the doctors very carefully and make the best decision at the time, but, it's like making the decision on the toes, you just have to give it a lot of thought."—Participant 7*

Some described asserting their wishes regarding amputation,

*"They wanted to cut my leg off above the knee, at first. And then I talked them out of that, I begged them not to do that. So they ended up doing more tests, and they found the artery that led to my lower shin area, right below the knee, that they thought might feed that area. So they agreed to just take below the knee. That was a win for me, I guess."—Participant 1*

Another told his surgeon that he was unwilling to lose his entire foot and opted to have an alternative procedure to improve his circulation. However, no patients were able to describe conversations about transmetatarsal versus transtibial amputations.

The lack of patient participation in amputation or amputation level decisions was coupled with a sense among many patients that amputation was inevitable due to ongoing health problems,

*"I wouldn't say that I was nonchalant about it, but it had been going on long enough, that you could almost see, that phrase, 'the writing on the wall'. You could see that there was a high probability that things would turn south and that you'd end up with an amputation out of this deal."*

*-Participant 4*

Several echoed the sentiment that "it had to be done,

*"No, he just told me that it's a procedure that has to be done. He said he wished he couldn't do it, but that it had to be done. That was just the bottom line."—Participant 3*

One discussed learning that amputation was a possibility when he was diagnosed with diabetes 20 years ago,

*"People I've known with diabetes, one day they're up and walking around, and then you see them the next time and they've lost a limb. And the next thing after that, they've lost two limbs."—Participant 12*

## Actively working towards recovery as response to a perceived loss of independence

While patients did not perceive that they participated in decisions around amputation or amputation level, many described an active role in the recovery process. Some talked about the importance of positive mindset,

*"I'm going to continue to fight. I'm not going to give up. I've got a bunch of other problems too, besides the amputation. I've been losing blood, I blacked out once. My blood count is always going down. I had (inaudible) and accepting that I am an amputee. A lot of people are in denial, ashamed of the wheelchair. I'm not ashamed of the wheelchair. I don't feel bad. I'm not doing a pity party; do you understand me ma'am?"—Participant 10*

Others discussed the importance of working hard to reach functional goals,

*"Look, this ain't stopping me, in fact, I plan on going faster, meaner and harder"—Participant 4*

Those goals were often related to desired levels of independence, including the ability to complete activities such as walking without a cane, getting from bed to a chair, or showering without assistance. When discussing functional goals, many participants expressed optimism and resolve. One participant was adamant that he walk without aid because he did not want to be seen as disabled,

*"And pretty soon, I'll be able to walk like normal. But at a shorter pace. That's what I have to deal with each day, and I do my exercise. Especially without the cane, because I want to get to walking where I don't need the cane, no kind of assistance from anything. Or anybody. It's like a personal thing with me. I don't want to be considered handicapped. That's what I'm dealing with each day. That's life. Deal with life and life'sturns. That's what I'm doing each day. I didn't think your toes meant so much, until they come up missing and you don't have them anymore.—Participant 8*

Many participants explicitly described their frustration with loss of independence, and in many cases loss of independence overshadowed the experience of losing a limb,

*"I've always been independent, so that was a big issue for me, sitting in a wheelchair at home, not being able to go anywhere except with somebody taking me. I haven't been social or out, pretty much for a year now. That's another thing that you don't think about, that everybody goes through. For me it's been over a year, for some it's less than that, but I'm sure for some people it's more than that. You can't drive a car, and if you've been independent your whole life, I worked until I was almost 70 years old, until this pain got so bad that I couldn't do it anymore"—Participant 1*

He further described, "*Not being able to be independent and having to depend on somebody is a big deal. It was for me anyway*" (*Participant 1*). Several participants focused on driving as a symbol of independence,

*Interviewer: What was that experience like for you?*

*Respondent: (Laughing) it was very depressing because I couldn't drive.*

*Interviewer: Oh. Can you tell me more about that?*

*Respondent: Oh my God, really? Can we skip that?."—Participant 9*

For others, loss of mobility and independence created other issues. Some were unable to work, which created financial problems,

*"They put a couple of grafts on it. But I couldn't take off from work to allow the grafts to take. You know, you're working, and nobody has any compassion for you. I just got skin grafts and staples in my foot, they don't want to hear that. They don't want to hear it. I initially got stitches, and then I think I got a staple, and then I had to take the staples out myself because I'm walking on it, I have to work! They put me in a foot cast, I had no choice. Bills don't go away, rent has got to be paid."—Participant 2*

### Experiencing amputation as a veteran

For some participants, being a Veteran affected how they interacted with their medical providers. One participant described treating his medical providers like officers,

*"I guess it goes back to being in the Marine Corps, the respect you give to, I treated my doctors like officers, that I had when I was in the Marine Corps, as far as 'yes sir', 'no sir'. I just put my trust in them and made the leap that they were going to do right by me and do what they needed to do to get me up and going."—Participant 6*

He described bringing an advocate to the appointments leading up to his amputation because he knew that he would not be able to ask questions about his options. He described how his advocate participated more actively in learning about the decision to amputate,

*"I had an advocate with me, she peppered him pretty good as far as certain questions about what we were going to face in the future, and they were very forthcoming with their answers. Basically, telling me a majority of it was up to me and how hard I work and what I want to accomplish. We were pleased with that. [. . .] She wanted more details as far as the Infectious Disease results, pain management, and things of that nature. Things that I wouldn't think to question."—Participant 6*

Other participants described connections with members of their medical teams who were also Veterans. One described having a surgeon who served in the air force,

*"I had asked him what it was like flying with the Wild Weasels, and I asked him if he ever missed being in aviation. There're not too many medical things that I could ask him, because he obviously knows what he's doing. The only thing we had in common was that I worked with the F18s in the Marine Corps, and a few other aircraft, so I was like, at least we had that in common."—Participant 4*

This common ground was a positive aspect of the patient's experience. Some also felt that Veteran healthcare providers were especially dedicated to Veterans due to prior service,

*"And then he went back to work, he just went back to work for the VA. I guess the only reason the man does it is because he cares, more than he needs the money. I don't imagine that he needs the money after being a retired surgeon. I had all confidence in him."—Participant 4*

He continued to describe his Veteran surgeon's wealth of experience with amputations with reference to service in the Vietnam War,

*"One of the surgeons had actually been in Vietnam and was talking about the differences between amputations then and now. From Vietnam until now, that's a good bit of experience in a particular area."- Participant 4*

For other patients, the feeling that amputation was common among Veterans was comforting,

*"I'm a veteran, I see a lot of veterans that have two amputations, I don't know how they did it, but if I had to, I would've."—Participant 10*

One described his amputation as "the cost of doing business" and discussed dangers he faced during military service that should have resulted in losing a limb or his life,

*"Before I was 20, I was probably like a lot of young fellas, I should've been dead 10 times before I got to be 20. And then, you know, I've had plenty of close misses in the military, so it wasn't anything out of violence that ended up taking me out, it was just a virus, or, whatever you want to call it, an infection in my leg that got into the bone. I guess I was blessed to have any day that I had with them."—Participant 4*

For other participants, interactions with fellow Veterans were instrumental to recovery,

*"But after [the amputation] happened, I'm on my own. Until I ran across [company name]/, which, my boss was a Vet and had a hip replacement, he worked it out for me."–Participant 2*

One participant wanted to help other Veterans recover from amputations,

*"I would love to teach the other guys. I might not even be able to teach them, I might just be able to be there and listen [. . .] I could push them around to their appointments and stuff. Particularly the old guys, some of them older guys are in their 70s and their wife is almost 70, and she's in no condition to be walking all over the place. My own little contribution back to society would be, I could lead by example, I'd say, 'hey, we've got to take care of everybody, each other particularly'. And it didn't matter if they were in WWII or if they were in Afghanistan last week."—Participant 4*

## Discussion

The goal of this study was to gain an understanding of Veteran lived experiences related to amputation due to CLTI and their involvement in SDM. Our findings suggest that most Veteran amputees did not feel that they were involved in SDM about their amputation, or amputation level. They felt that amputation was necessary and deferred to their medical teams' suggestions. While most patients did not feel able to participate in amputation decisions, they described active participation in the recovery process.

It was striking that patients did not perceive a decision around amputation level. Recognizing the need for a decision is key to SDM [21], and if patients are not presented with a decision around amputation level they are not able to communicate with their providers about values that are important to them, such as leisure activities, ability to work, home configuration, or healing time. Some patients expressed frustration that they were not involved in decision making. Prior research finds that lack of information is a barrier to SDM [15,16], but our results suggest that lack of perceived decision is a more proximate barrier.

Presenting amputation level as a choice may be especially important to diabetic patients. Our results suggest that for many participants, the ongoing nature of their disease and its many complications made amputation feel inevitable. Many patients with diabetes have additional chronic conditions [22,23] and research shows that diabetic patients tend to perceive

that they are at high risk for complications like lower limb amputations [24]. It is possible that this sense of inevitability impacts how patients interact with their medical teams to understand how different amputation levels may correlate with their goals or impact quality of life after amputation. Evidence exists that risks for mortality [25], reamputation [26], and achieving an independent level of mobility [27] may differ by amputation level depending on the composite of a patients individual risk factors; therefore, understanding the risks and benefits of each amputation level may aid the discussion around the amputation level that best aligns with the patient's individual priorities and goals.

The finding that being a Veteran was central to the lived experience for many participants has important implications for SDM. Some Veterans stated that because of their military training they felt less able to interact with and question their medical providers about their treatment. In a previous study assessing participation in SDM among Veteran patients, Rodriguez et al [28] found that a majority of participants took a passive role in healthcare decision making, but many would have preferred a more active role. Naik et al. [29] similarly found that many Veterans have poor health literacy, and that poor health literacy was a barrier to patient participation in SDM. While our findings echo those in previous work, they add further context and suggest that deference to authority, or authority bias, may also be a barrier to SDM in Veteran papulations. Previous work shows that deference to authority inhibits SDM between medical providers and older adults [30]. Together, these findings suggest that medical providers attempting to engage patients in SDM should be sensitive to their authority and situate conversations with patients in the context of a power differential.

All of the participants in this study were male Veterans and further research is needed to understand how the lived experience of females or non-Veterans regarding lower limb amputees may be similar or different.

## Conclusions

Our findings suggest that the lived experience of lower limb amputees does not involve a decision around whether to amputate or amputation level. Further, we find that Veteran patients may face unique barriers to participating in SDM. These findings have implications for involving Veteran patients in amputation level decisions. Patients should not only be made aware that there is a decision around amputation level, but they may also need advocates or additional support when interacting with medical teams. This information will be invaluable in designing patient decision aids or decision support tools to assist patients in the amputation level shared decision-making process.

## Supporting information

**S1 File.**
(DOCX)

## Acknowledgments

The authors would like to gratefully acknowledge study participants for sharing their time and perspectives.

## Government employee

All authors work for the United States Government (Department of Veterans Affairs).

## Author Contributions

**Conceptualization:** George Sayre, Alison Henderson, Daniel Norvell, Aaron P. Turner, Joseph Czerniecki.

**Formal analysis:** Chelsea Leonard, Sienna Williams.

**Funding acquisition:** Alison Henderson, Daniel Norvell, Aaron P. Turner, Joseph Czerniecki.

**Methodology:** Chelsea Leonard, George Sayre.

**Project administration:** Alison Henderson.

**Writing – original draft:** Chelsea Leonard.

**Writing – review & editing:** George Sayre, Sienna Williams, Alison Henderson, Daniel Norvell, Aaron P. Turner, Joseph Czerniecki.

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
