## [Decision Letter · Decision Letter 0]

19 Dec 2021

PONE-D-21-23145Understanding the experience of Veterans who require dysvascular lower limb amputation in the Veterans Health AdministrationPLOS ONE

Dear Dr. Leonard,

Thank you for submitting your manuscript to PLOS ONE. After careful consideration, we feel that it has merit but does not fully meet PLOS ONE’s publication criteria as it currently stands. Therefore, we invite you to submit a revised version of the manuscript that addresses the points raised during the review process.

We look forward to receiving your revised manuscript.

Kind regards,

Yih-Kuen Jan, PhD, University of Illinois at Urbana-Champaign

2. Thank you for including your ethics statement:  "This study was approved by the local facility institutional review board: MIRB 01700; IRBNet # 1587998-12

“This material is based upon work supported by the US Department of Veterans Affairs, Office of Research and Development, Rehabilitation Research and Development Grant number 1 I01 RX002960-01.”

“This study was made possible by funding from the VA Office of Research and Development. The authors would like to gratefully acknowledge study participants for sharing their time and perspectives.”

“This material is based upon work supported by the US Department of Veterans Affairs, Office of Research and Development, Rehabilitation Research and Development Grant number 1 I01 RX002960-01.”

6. We note that you have indicated that data from this study are available upon request. PLOS only allows data to be available upon request if there are legal or ethical restrictions on sharing data publicly. For more information on unacceptable data access restrictions, please see http://journals.plos.org/plosone/s/data-availability#loc-unacceptable-data-access-restrictions.

Reviewers' comments:

Reviewer's Responses to Questions

**Comments to the Author**

1. Is the manuscript technically sound, and do the data support the conclusions?

Reviewer #1: Yes

Reviewer #2: Yes

2. Has the statistical analysis been performed appropriately and rigorously? 

Reviewer #1: Yes

Reviewer #2: N/A

3. Have the authors made all data underlying the findings in their manuscript fully available?

Reviewer #1: Yes

Reviewer #2: Yes

4. Is the manuscript presented in an intelligible fashion and written in standard English?

Reviewer #1: Yes

Reviewer #2: Yes

5. Review Comments to the Author

Reviewer #1: Surgeons performing the amputation may not have all the information as to rehab/prosthetic options. Would have been interesting to assess the surgeons' level of knowledge of life after surgery, because that may have affected their willingness to engage in meaningful discussions prior to amputation.

Reviewer #2: This study is trying to explore the experience of Veterans during the amputation in VHA. A good engagement can usually promote the participation of patients with amputation. The insufficient shared decision making (SDM) during amputation is found in this study. To improve the benefits of amputees, this find is an important step for the development of a decision support tool and can help them engage the involvement. Although only twelve patients were recruited in this study, the results still showed an implication for an obstacle for the patients with CLTI to participate in the treatment of their own. This study is qualitative research and reveals some important points for further studies. However, more investigation should be done in the future to support the finding from this research due to the small sample size and limited sampling.

There are several minor issues that need to be addressed before publication.

1. The amputation usually changes the whole lifestyle of the people, sometimes it also makes the amputees under negative emotions. Most people with amputation feel sadness, madness, or other strong and negative emotions. Is SDM or involvement affected by undergoing these emotions? Could someone who is more optimistic feel they were involved more (or less) before surgery (in contrast, the subjects under sadness might feel more helpless or out of control)?

Did you observe any correlation between their responses or feels and the emotions of the subjects?

2. The authors mentioned that most subjects with diabetes perceive the amputation might happen to them. Most subjects in this study believe that the medical team would make the best choice and they don’t need to, or they couldn’t provide any personal thoughts on the discussion. Is there any possibility that the surgery should be done in rush time so made the medical team had no time to let the patients be involved in the decision-making? During the interview, had the subjects in this study ever mentioned or suggested that the debate on the amputation level could be done earlier? Would it be better if discuss with the patients with CLTI about the possible amputation level when the condition wasn’t going so badly?

As the authors indicate in the study design, this research is a first but essential step before more extensive research to develop the patient decision aid and decision support tool.

6. PLOS authors have the option to publish the peer review history of their article (what does this mean?). If published, this will include your full peer review and any attached files.

Reviewer #1: No

Reviewer #2: **Yes: **Mo, Pu-Chun

---

## [Author Response · Author response to Decision Letter 0]

1 Feb 2022

Please see attached reviewer response document for a formatted table of author responses to requested edits. 

Please ensure that your manuscript meets PLOS ONE's style requirements, including those for file naming. The PLOS ONE style templates can be found at Thank you for this feedback. 

We have updated the formatting of the manuscript.

2. Thank you for including your ethics statement: "This study was approved by the local facility institutional review board: MIRB 01700; IRBNet # 1587998-12 

Thank you for this comment, we have updated our ethics statement to include the name "Human Research Protection Program, VA Puget Sound Health Care System"

We added this statement on the submission form.

“This material is based upon work supported by the US Department of Veterans Affairs, Office of Research and Development, Rehabilitation Research and Development Grant number 1 I01 RX002960-01.” 

Thank you. The financial disclosure should read as follows: Role of Funder: This material is based upon work supported by the US Department of Veterans Affairs, Office of Research and Development, Rehabilitation Research and Development Grant number 1 I01 RX002960-01. The funders had no role in study design, data collection and analysis, decision to publish, or preparation of the manuscript.

Thank you. We included this in the cover letter. 

“This study was made possible by funding from the VA Office of Research and Development. The authors would like to gratefully acknowledge study participants for sharing their time and perspectives.” 

Thank you. We have removed funding information from the Acknowledgements. 

Thank you. We have removed funding related text from the manuscript.

“This material is based upon work supported by the US Department of Veterans Affairs, Office of Research and Development, Rehabilitation Research and Development Grant number 1 I01 RX002960-01.” 

Thank you. We have included the following amended statement in the cover letter: This material is based upon work supported by the US Department of Veterans Affairs, Office of Research and Development, Rehabilitation Research and Development Grant number 1 I01 RX002960-01. The funders had no role in study design, data collection and analysis, decision to publish, or preparation of the manuscript.

Thank you. We have included the following in our Data Availability Statement: This paper reports on a qualitative study and excerpts of appropriate data are shared within the manuscript. We are unable to provide full de-identified interview transcripts as this was not discussed with participants during the consent process. 

We are unable to upload additional data, as per the data availability statement above. We meet the criteria for minimal dataset for qualitative data by sharing data excerpts within the manuscript.

6. We note that you have indicated that data from this study are available upon request. PLOS only allows data to be available upon request if there are legal or ethical restrictions on sharing data publicly. For more information on unacceptable data access restrictions, please see http://journals.plos.org/plosone/s/data-availability#loc-unacceptable-data-access-restrictions.

Thank you. We revised our data availability statement (please see above).

a) If there are ethical or legal restrictions on sharing a de-identified data set, please explain them in detail (e.g., data contain potentially sensitive information, data are owned by a third-party organization, etc.) and who has imposed them (e.g., an ethics committee). Please also provide contact information for a data access committee, ethics committee, or other institutional body to which data requests may be sent. Thank you. We have added this information to our cover letter. 

Thank you. We have reviewed our references and are not making any changes.

Reviewers' comments: 

Comments to the Author

5. Review Comments to the Author

Reviewer #1: Surgeons performing the amputation may not have all the information as to rehab/prosthetic options. Would have been interesting to assess the surgeons' level of knowledge of life after surgery, because that may have affected their willingness to engage in meaningful discussions prior to amputation. 

Thank you for this observation. We agree that it would be interesting to assess surgeons' level of knowledge of life after amputation and will address that question in future qualitative work. The goal of the current study was to understand patient experiences in their own words, and we did not talk to surgeons in this portion of the study. 

Reviewer #2: This study is trying to explore the experience of Veterans during the amputation in VHA. A good engagement can usually promote the participation of patients with amputation. The insufficient shared decision making (SDM) during amputation is found in this study. To improve the benefits of amputees, this find is an important step for the development of a decision support tool and can help them engage the involvement. Although only twelve patients were recruited in this study, the results still showed an implication for an obstacle for the patients with CLTI to participate in the treatment of their own. This study is qualitative research and reveals some important points for further studies. However, more investigation should be done in the future to support the finding from this research due to the small sample size and limited sampling. 

Thank you for this comment. We agree that the sample size of the current study was small, even for a qualitative study. We selected a small sample because this was an in-depth phenomenological study intended to inform a future larger qualitative study, and we are confident that we reached data saturation within our small sample. In the next phase of this study, we will follow up on our findings with a larger sample of patients. 

There are several minor issues that need to be addressed before publication. 

1. The amputation usually changes the whole lifestyle of the people, sometimes it also makes the amputees under negative emotions. Most people with amputation feel sadness, madness, or other strong and negative emotions. Is SDM or involvement affected by undergoing these emotions? Could someone who is more optimistic feel they were involved more (or less) before surgery (in contrast, the subjects under sadness might feel more helpless or out of control)? Thank you for this question. We are unable to address this question with the current dataset but hope to explore these ideas in another set of interviews with both patients and surgeons to learn how emotion affects their participation in SDM. 

Did you observe any correlation between their responses or feels and the emotions of the subjects? 

This is a good question. I have revised the second theme, "Actively working towards recovery as response to a perceived loss of independence" to reflect participants' emotional responses as this is where the analysts noted emotional response in the interviews. 

2. The authors mentioned that most subjects with diabetes perceive the amputation might happen to them. Most subjects in this study believe that the medical team would make the best choice and they don’t need to, or they couldn’t provide any personal thoughts on the discussion. Is there any possibility that the surgery should be done in rush time so made the medical team had no time to let the patients be involved in the decision-making? During the interview, had the subjects in this study ever mentioned or suggested that the debate on the amputation level could be done earlier? Would it be better if discuss with the patients with CLTI about the possible amputation level when the condition wasn’t going so badly? 

This is a good observation. As shown in Table 2, most of the participants in this study received amputations in emergent settings, and it is very likely that this affected their experience with lack of SDM. In the subsequent stage of this study, we asked both patients and their surgeon's when amputation level conversations should take place. The goal of the current study was to understand participants' experiences more broadly, and the timing of amputation level conversations was not emergent in the interview data. 

As the authors indicate in the study design, this research is a first but essential step before more extensive research to develop the patient decision aid and decision support tool.

---

## [Decision Letter · Decision Letter 1]

7 Mar 2022

Understanding the experience of Veterans who require lower limb amputation in the Veterans Health Administration

PONE-D-21-23145R1

Dear Dr. Leonard,

We’re pleased to inform you that your manuscript has been judged scientifically suitable for publication and will be formally accepted for publication once it meets all outstanding technical requirements.

Kind regards,

Yih-Kuen Jan, PhD

Academic Editor

PLOS ONE

Additional Editor Comments (optional):

Reviewers' comments:

Reviewer's Responses to Questions

**Comments to the Author**

1. If the authors have adequately addressed your comments raised in a previous round of review and you feel that this manuscript is now acceptable for publication, you may indicate that here to bypass the “Comments to the Author” section, enter your conflict of interest statement in the “Confidential to Editor” section, and submit your "Accept" recommendation.

Reviewer #2: All comments have been addressed

2. Is the manuscript technically sound, and do the data support the conclusions?

Reviewer #2: Yes

3. Has the statistical analysis been performed appropriately and rigorously? 

Reviewer #2: N/A

4. Have the authors made all data underlying the findings in their manuscript fully available?

Reviewer #2: Yes

5. Is the manuscript presented in an intelligible fashion and written in standard English?

Reviewer #2: Yes

6. Review Comments to the Author

Reviewer #2: All the comments were adequately addressed. This article is necessary to build up a good decision support tool and shows that the provided system of helping Veterans during the amputation is insufficient. Expect further studies to be done soon.

7. PLOS authors have the option to publish the peer review history of their article (what does this mean?). If published, this will include your full peer review and any attached files.

Reviewer #2: No

---

## [Editor Report · Acceptance letter]

10 Mar 2022

PONE-D-21-23145R1 

Understanding the experience of Veterans who require lower limb amputation in the Veterans Health Administration 

Dear Dr. Leonard:

I'm pleased to inform you that your manuscript has been deemed suitable for publication in PLOS ONE. Congratulations! Your manuscript is now with our production department. 

Kind regards, 

on behalf of

Dr. Yih-Kuen Jan 

Academic Editor

PLOS ONE